# Rasagiline Exerts Neuroprotection towards Oxygen–Glucose-Deprivation/Reoxygenation-Induced GAPDH-Mediated Cell Death by Activating Akt/Nrf2 Signaling

**DOI:** 10.3390/biomedicines12071592

**Published:** 2024-07-17

**Authors:** Shimon Lecht, Adi Lahiani, Michal Klazas, Majdi Saleem Naamneh, Limor Rubin, Jiayi Dong, Wenhua Zheng, Philip Lazarovici

**Affiliations:** 1School of Pharmacy Institute for Drug Research, Faculty of Medicine, The Hebrew University of Jerusalem, Jerusalem 9112002, Israel; 2Allergy and Clinical Immunology Unit, Department of Medicine, Hadassah-Hebrew University Medical Center, Jerusalem 9112001, Israel; 3Center of Reproduction, Development & Aging, Faculty of Health Sciences, University of Macau, Taipa, Macau 999078, China

**Keywords:** akt, cell death, GAPDH, ischemia-like insult, neuroprotection, Nrf2, phosphorylation, rasagiline, ROS, PC 12 neuronal model, α-synuclein, Parkinson’s disease, stroke

## Abstract

Rasagiline (Azilect^®^) is a selective monoamine oxidase B (MAO-B) inhibitor that provides symptomatic benefits in Parkinson’s disease (PD) treatment and has been found to exert preclinical neuroprotective effects. Here, we investigated the neuroprotective signaling pathways of acute rasagiline treatment for 22 h in PC12 neuronal cultures exposed to oxygen–glucose deprivation (OGD) for 4 h, followed by 18 h of reoxygenation (R), causing 40% aponecrotic cell death. In this study, 3–10 µM rasagiline induced dose-dependent neuroprotection of 20–80%, reduced the production of the neurotoxic reactive oxygen species by 15%, and reduced the nuclear translocation of glyceraldehyde-3-phosphate dehydrogenase (GAPDH) by 75–90%. In addition, 10 µM rasagiline increased protein kinase B (Akt) phosphorylation by 50% and decreased the protein expression of the ischemia-induced α-synuclein protein by 50% in correlation with the neuroprotective effect. Treatment with 1–5 µM rasagiline induced nuclear shuttling of transcription factor Nrf2 by 40–90% and increased the mRNA levels of the antioxidant enzymes *heme oxygenase-1*, *(NAD (P) H- quinone dehydrogenase*, and *catalase* by 1.8–2.0-fold compared to OGD/R insult. These results indicate that rasagiline provides neuroprotection to the ischemic neuronal cultures through the inhibition of α-synuclein and GAPDH-mediated aponecrotic cell death, as well as via mitochondrial protection, by increasing mitochondria-specific antioxidant enzymes through a mechanism involving the Akt/Nrf2 redox-signaling pathway. These findings may be exploited for neuroprotective drug development in PD and stroke therapy.

## 1. Introduction

Parkinson’s disease (PD) is a complex neurodegenerative disease of the dopaminergic neurons located in the brain substantia nigra pars compacta. It is characterized by progressive dopaminergic neuron degeneration and Lewy body formation [1]. This degeneration and the decline in the functional presynaptic dopaminergic activity in the PD brain leads to a dramatic decrease in dopamine (DA) levels [2], triggering the typical clinical symptoms of bradykinesia, rigidity, resting tremors, postural instability, etc., as well as anatomical changes in brain tissue, as shown by imaging techniques [3]. The etiology of PD is likely multifactorial and involves interplay among aging, genetic susceptibility, and environmental factors [4]. Although the pathophysiologic mechanism of PD remains unknown, many studies have shown that oxidative stress plays an important role in the cell death of dopaminergic neurons [5,6,7]. Dopamine can undergo oxidation to induce the production of reactive oxygen species (ROS) and electrophilic quinone molecules [8], explaining the susceptibility of dopaminergic neurons to oxidative and electrophilic stress [9]. Moreover, targeting oxidative stress has received widespread recognition considering that ROS play key roles in normal brain function and pathology in the context of many neurological diseases, including stroke, traumatic brain injury, etc. [10].

Glyceraldehyde-3-phosphate dehydrogenase (GAPDH), besides its function in converting glyceraldehyde 3-phosphate to 1, 3-biphosphoglycerate during glycolysis, is implicated in several non-metabolic processes, including transcriptional activation and cell apoptosis [11]. GAPDH is overexpressed and accumulates in the nucleus during apoptosis in response to oxidative stress, which is triggered by its nuclear translocation induced by a variety of insults in diverse cell types, including PC12 cells [12,13]. The phosphatidylinositol 3-kinase-protein kinase B (PI3K) (-) Akt signaling pathway regulates signal transduction and biological processes such as cell proliferation, apoptosis, and metabolism; regulates neurotoxicity; and mediates the survival of neurons [14]. Many findings indicate that Akt phosphorylates several targets, suggesting that it modulates neuronal cell death by both impinging on cytoplasmic cell death machinery and regulating nuclear proteins [15]. One mechanism by which Akt kinase suppresses GAPDH-mediated apoptosis is via phosphorylating GAPDH at threonine 237 and decreasing its nuclear translocation [16].

The neuron maintains a robust antioxidant defense mechanism consisting of several neuroprotective genes and enzymes whose expression is controlled by antioxidant response element (ARE), which further depends on the activation of the nuclear factor erythroid 2-related factor 2 (Nrf2). In response to oxidative or electrophilic stress, this redox-sensitive transcription factor, Nrf2, binds to ARE and rescues the cells from oxidative stress via the induction of ARE-mediated expression of phase II detoxifying antioxidant enzymes, including NAD(P)H-quinone oxidoreductase1 (NQO1), heme oxygenase-1 (HO-1), catalase, etc. [17]. The nuclear factor erythroid 2-related factor 2 (Nrf2)-Keap1 orchestrates the antioxidant response in the brain by promoting the expression of several antioxidant enzymes that exert neuroprotective effects against oxidative damage and mitochondrial impairment [18,19]. Activation of Nrf2 by drugs or genetic manipulations is demonstrated to alleviate PD induced by agents such as MPP+, rotenone, and H_2_O_2_, as well as genetic factors that can protect neurons in vitro and in vivo against DA-related neurotoxicity [20]. Hence, drugs that can induce the upregulation of ARE-mediated expression of phase II detoxifying antioxidant enzymes via the Nrf2/ARE pathway would be a promising approach for neuroprotection in PD and stroke [21,22]. 

α-synuclein is a neuronal synaptic protein that is a major component of Lewy bodies, neuronal cytoplasmic inclusions of aggregated proteins, which are typical biomarkers of idiopathic and familial forms of PD [23]. Extensive evidence shows that α-synuclein is neurotoxic and that it is implicated in the pathophysiology of PD [24] and stroke [25]. Recent studies have also provided evidence of its relation to neuroprotection, as it can inhibit apoptosis in response to various pro-apoptotic signals [26]. α-synuclein is expressed and selectively up-regulated in response to nerve growth factor treatment in PC12 cell cultures [27]. Despite conflicting data that still exist, understanding the homeostatic balance of α-synuclein expression and activity is important and calls for further investigations regarding the physiological role of this neuronal protein during PD and stroke and upon treatment with anti-Parkinsonian drugs.

Rasagiline (Azilect^®^, Teva Co., Tel Aviv, Israel) is a selective, irreversible monoamine oxidase type-B (MAO-B) second-generation inhibitor that is clinically used in PD patients [28]. Rasagiline has been demonstrated to be neuroprotective in PD and oxidative stress model systems by preventing the formation of ROS derived from the oxidation of dopamine by MAO-B and via an anti-apoptotic action, which appears to be independent of MAO-B inhibition and related to its pharmacophore N-propargyl moiety [29,30,31]. Furthermore, chronic treatment with rasagiline induces indirect antioxidant activity by enhancing the expression of anti-oxidative enzymes such as SOD1, SOD2, and catalase [32]. 

To investigate the cellular and molecular neuroprotective mechanisms of rasagiline under ischemia, we used pheochromocytoma PC12 dopaminergic neuronal cell cultures that express MAO-B and exposed them to an oxygen–glucose deprivation (OGD) and reoxygenation (R) protocol (OGD/R), representing a well-established in vitro model of the ischemic insult [33,34]. We hypothesized that rasagiline conferred neuroprotection to PC12 cell cultures exposed to OGD/R oxidative stress insult by decreasing the nuclear translocation of glyceraldehyde-3-phosphate dehydrogenase and α-synuclein protein levels, and by activating the Akt/Nrf2 redox-signaling pathway. Therefore, we investigated whether rasagiline induced the Nrf2-ARE signaling pathway, leading to an increase in ARE-dependent antioxidant enzyme mRNA expression in relation to its neuroprotective effect in PC12 cell cultures exposed to OGD/R. We also explored the temporal effects of Akt activation and GAPDH nuclear localization in relation to rasagiline-induced neuroprotection. The findings provide a mechanistic proposal for rasagiline’s neuroprotective effects towards ischemic-like insults that may add to its future drug development and repurposing for some aspects of PD and stroke therapy.

## 2. Materials and Methods

### 2.1. Materials

Rasagiline mesylate was supplied by Teva pharmaceutics company, Israel Ltd., and stored as a stock solution of 10 mM (5 mg dissolved in 1.87 mL DMSO). Tempol, (4-hydroxy-2, 2, 6, 6-tetramethylpiperidine-d17-1-oxyl), a membrane-permeable radical scavenger, and LY294002, a cell-permeable, potent, and reversible specific inhibitor of PI3-kinase, were purchased from Sigma-Aldrich Chemical Co. (Jerusalem, Israel) and used as received. The collagen type Ι was purchased from BD Biosciences (Bedford, MA, USA). MitoSOX™ Red mitochondrial superoxide probe (M36008) was purchased from Invitrogen Molecular Probes (Eugene, OR, USA). Polyvinylidene difluoride (PVDF) membranes were obtained from Invitrogen (Invitrogen Corporation, Carlsbad, CA, USA). mNGF, HPLC-grade, was purchased from Alomone Labs (Jerusalem, Israel). Dulbecco’s modified Eagle’s medium (DMEM), fetal calf serum, donor horse serum, penicillin, and streptomycin were all purchased from Biological Industries (Beit Haemek, Afula, Israel).

### 2.2. PC12 Cell Cultures 

Rat pheochromocytoma PC12 cells were propagated in 25 cm^2^ flasks in growth medium composed of Dulbecco’s modified Eagle’s medium (DMEM) supplemented with 7% fetal calf serum (FCS), 7% horse serum (HS), 10,000 U/mL penicillin, and 100 μg/mL streptomycin. The medium was replaced every second day, and cells were grown at 37 °C in a humidified atmosphere of 6% CO_2_ [35]. To differentiate the cells, PC12 cell cultures grown on 6-well plates (0.5 × 10^5^ cells/well) and glass coverslips (1 × 10^4^ cells/coverslip) were treated with 50 ng/mL mouse β-NGF in DMEM with serum and antibiotics every 48 h for a period of 8–10 days before being subjected to the ischemic insult.

### 2.3. Ischemic Insult Protocol 

PC12 cells (1 × 10^5^ cells/well) were applied to 12-well plates or glass coverslips in 6-well plates, pre-coated with 200 μg/mL collagen type-I, and grown for 2 days. On the day of the experiment, the cell medium was replaced with glucose-free DMEM (hypoglycemic insult), and the cultures were introduced into an ischemic chamber with an oxygen level below 1% (hypoxic insult) for 4 h at 37 °C under oxygen and glucose deprivation (OGD), as previously described [35]. To mimic in vivo reperfusion conditions, at the end of the OGD insult, 4.5 mg/mL glucose was added, and cultures were incubated for an additional 18 h under normoxic conditions (reoxygenation, R) to complete the ischemic insult. Operationally, the ischemic insult consisted of a combination of both OGD and reperfusion phases (OGD/R). Control cultures were maintained under regular atmospheric conditions (normoxia) in the presence of 6% CO_2_ with high-glucose DMEM. The addition of 10 μM rasagiline was performed before OGD and maintained during the reperfusion (22 h treatment). At the end of the reperfusion phase, cell death was measured using the LDH release method [36] at 340 nm with a spectrofluorometer (TECAN, SPECTRA Fluor PLUS, Salzburg, Austria). LDH release was expressed as the percent of total LDH released into the medium upon subtracting the basal values of LDH release. The neuroprotective effect, defined as the percent decrease in LDH release in the presence of rasagiline, was normalized to untreated ischemic cultures. 

### 2.4. Measurement of Mitochondrial Reactive Oxygen Species (ROS) Levels

Mitochondrial superoxide in live cells was detected using the MitoSOX™ Red mitochondrial superoxide probe. PC12 cells were cultured in 6-well plates (5 × 10^5^ cells/well) and exposed to the OGD/R in the presence or absence of 10 µM rasagiline or 1 mM tempol, which was used as a positive control. The plates were then centrifuged at 1000 rpm for 5 min, and the medium was removed. The cell cultures were incubated with 5 mM MitoSOX™ reagent dissolved in dimethyl sulfoxide working solution in Hanks’ Balanced Salt Solution (HBSS/Ca^2+^/Mg^2+^) (Bet Haemek, Afula, Israel) in the dark at 37 °C for 10 min. The cells were washed gently three times with HBSS/Ca^2+^/Mg^2+^, and the fluorescence intensity of the cells was detected by flow cytometry. For each sample, 10^5^–10^6^ events were acquired with a BD LSRII flow cytometer (BD Biosciences). ROS levels were expressed as the mean fluorescence intensity of 10^5^ cells/sample, calculated by the FCS Express 6 Plus analysis software (De Novo Software Version 6) [37]. 

### 2.5. Nuclear Protein Extraction

PC12 cells (2 × 10^6^), either naïve or knocked-down for Nrf2, were grown for 24 h in a 10 cm dish under normoxia or exposed to the OGD/R insult, and then rasagiline was added into the medium of each dish. At the end of the ischemic insult, the cell cultures were harvested by scraping. The cell pellets were washed by PBS and then re-suspended in 50 μL of extraction buffer I (10 mM KCl, 10 mM HEPES, 1.5 mM MgCl_2_; with 0.5 mM DTT and 0.1% NP-40 freshly added before use) and incubated on ice for 15 min, then centrifuged at 6000 rpm and 4 °C for 10 min. The pellets containing the nuclei were re-suspended in 50 μL of nuclear extraction buffer II (20 mM HEPES (pH 7.9), 25% glycerol, 420 mM NaCl, 0.2 mM EDTA, 1.5 mM MgCl_2,_ with 0.5 mM DTT, and 0.5 mM phenylmethylsulfonyl fluoride (PMSF)); incubated on ice for 15 min; and then centrifuged at 3000 rpm for 10 min at 4 °C. The supernatant containing nuclear proteins was diluted with 50 μL of buffer III (20 mM HEPES (pH 7.9), 20% glycerol, 50 mM KCl, 0.2 mM EDTA with 0.5 mM DTT, and 0.5 mM PMSF added freshly) and stored at −80 °C [13]. A BCA protein assay kit (PIERCE) was used to determine the protein concentration.

### 2.6. Western Blotting

At the end of the ischemic insult, PC12 cell cultures were collected by centrifugation at 2000 rpm for 5 min and washed with 1 mL of fresh PBS. Thereafter, the cells were suspended in lysis buffer (20 mM HEPES pH 7.4 containing 2 mM EGTA, 50 mM glycerol phosphate, 1% Triton X-100,10% glycerol, 1 mM dithiothreitol, 1 mM PMSF, 10 mg/mL leupeptin, 10 mg/mL aprotinin, 1 mM Na_3_VO_4_, and 5 mM NaF) and incubated for 10 min on ice. The lysed samples were sonicated for 5 sec and centrifuged at 15,000 rpm for 15 min to separate the protein extract from the pellet. The protein concentration was determined according to the BCA protein assay kit. Samples containing 30 μg of cell protein were boiled for 5 min in SDS sample buffer and separated by 10%, 12%, or 15% sodium dodecyl sulfate polyacrylamide gel electrophoresis (SDS-PAGE). The protein bands were subsequently transferred to PVDF membranes. The membranes were blocked at room temperature for 1 h using TBST (20 mM Tris-HCl, 137 mM NaCl, pH 7.5, and 0.1% Tween-20) containing 5% milk. After washing with TBST, the membranes were incubated overnight at 4 °C with the primary antibodies and then washed with TBST. The primary antibodies employed were directed towards cleaved caspase-3 (Santa Cruz Biotechnology, Dallas, TX, USA, 1:100); cleaved PARP (Santa Cruz Biotechnology, 1:500); Bcl-2 (Santa Cruz Biotechnology, 1:200); phospho-Akt (Ser 473, Cell Signaling Technology, Danvers, MA, USA, 1:2000); pan-Akt (Cell Signaling Technology, 1:1000); α-synuclein (BD Transduction Laboratories, 1:500); nuclear factor erythroid 2-related factor 2 (Cell Signaling Technology, 1:500); NAD(P)H: quinone oxidoreductase 1 (Calbiochem, San Diego, CA, USA, NQO1; 1:1000); β-actin (Santa Cruz Biotechnology, 1:3000); tubulin (anti-βIII tubulin, Chemicon–Millipore, Burlington, MA, USA, 1:200); and lamin B (Santa Cruz Biotechnology, 1:500). In the next step, the membranes were washed three times with TBST for 10 min before treatment for 2 h with the secondary antibody−HRP conjugates of either sheep anti-mouse (1:3000), donkey anti-rabbit (1:3000) (Jackson Immunoresearch, West Grove, PA, USA), or rabbit antibodies to goat IgG (1:3000, Santa Cruz Biotechnology). For Akt bands, the densitometric values were obtained for the phosphorylated and pan antibodies. The background of each film was subtracted, and the relative density of the band of phosphorylated Akt was divided by the density of the respective band of the pan-protein. The data are presented as the percentage of phosphorylated Akt from the total pan-Akt. Specific antibody binding was measured by enhanced chemiluminescence (ECL), visualized using Image Lab Software 5.1 (Bio-Rad Laboratories, Hercules, CA, USA), and quantified [38].

### 2.7. Knockdown of Nrf 2 Gene Expression Using siRNA

The commercial Amine Transfection Agent protocol was followed (Ambion, Austin, TX, USA). SiRNASiPORT™ Amine–Polyamine-Based Transfection Agent is a propriety blend of polyamines formulated for transfection of small RNAs. The reagent functions by complexing with RNA and facilitating its transfer into the cells. In brief, 10 nM of Silencer^®^ Select Predesigned mixture of siRNAs (Ambion, Austin, TX, USA) was directed against Nrf-2α and Nrf-2β (NM_031789.3). The sense sequences of Nrf-2α and Nrf-2β siRNAs were 5′-GAAUUCAGCAUGACCGAUAtt-3′ and 5′-GGUGGAACUUUUAAUCAAAtt-3′, respectively [39]. Nrf2 siRNAs or Silencer^®^ Select Negative Control (scrambled) siRNA (Ambion), diluted in Gibco™ OPTI-MEM^®^I (Invitrogen, New York, NY, USA), was combined with siPORT™ Amine Transfection Agent diluted in Gibco™ OPTI-MEM^®^I. This mixture was added to a 6-well NUNC™ TC plate containing 2 × 10^5^ PC12 cells per well. The cell cultures were subjected to daily treatment with rasagiline, starting 12 h after treatment with the siRNAs and lasting up to 6 days post-transfection, followed by RNA extraction and RT-PCR analyses. Following 72 h of incubation, a new transfection reagent containing Nrf2 or scrambled siRNA was added to the appropriate groups, an approach previously used by us to knock down the tuftelin gene [40].

### 2.8. RNA Extraction and RT-PCR

Total RNA from the PC12 cell cultures was extracted using Trizol (Invitrogen Corp., New York, NY, USA), and the total RNA concentration was determined using a NanoDrop ND-1000 spectrophotometer (Nano-Drop Technologies, Wilmington, DE, USA). Thereafter, 1 µg of total RNA of each sample was subjected to reverse transcription according to the manufacturer’s protocol using a High-Capacity cDNA Reverse Transcription Kit (Applied Biosystems^®^, Foster City, CA, USA), and real-time quantitative PCR was performed using an Applied Biosystems 7300 Real-Time-PCR System with Applied Biosystems TaqMan^®^ AOD (Assays-On-Demand) and Master Mixes (TaqMan^®^, Gene Expression Assays). Briefly, 2 µL cDNA was mixed with GoTaq Green Master Mix (Promega. Madison, WI, USA) and with primers for NQO1 (NM-017000.3; amplicon length 75 bp), heme oxygenase 1 (HO-1; NM- 20012580; amplicon length 93 bp), glutathione peroxidase 2 (GPx2; NM-183403.2; amplicon length 74 bp), catalase (NM-012520.2; amplicon length 76 bp), and GAPDH (internal control;NM-017008.4; amplicon length 87 bp). The primers for these genes were: NQO1, 50-TTCTGTGGCTTCCAGGTCTT-30 (forward) and 50-AGGCTGCTTGGAGCAAAATA-30 (reverse); HO-1, 50-CTTTCAGAAGGGTCAGGTGTC-30 (forward) and 50-TGCTTGTTTCGCTCTATCTCC-30 (reverse); catalase, 50-AAATGCTTCAGGGCCGCCTT -30 (forward) and 50-GTAGGGACAGTTCACAGGTA-30 (reverse); and GAPDH, 50-GTATTGGGCGCCTGGTCACC-30 (forward) and 50-CGCTCCTGGAAGATGGTGATGG-30 (reverse). The PCR products were also electrophoresed in 2% agarose gel and visualized using ethidium bromide. The PCR parameters were as follows: initial denaturation at 95 °C for 10 min, followed by 40 cycles of denaturation at 95 °C for 10 s, annealing at 60 °C for 15 s, and elongation at 72 °C for 20 s. The mRNA levels were expressed as the relative copy number of each target mRNA to GAPDH for each sample, and the cycle threshold of the control group was normalized to 1 [40].

### 2.9. Immunofluorescence and Confocal Microscopy

PC12 cell cultures grown on microscope glass coverslips under normoxia or exposed to the OGD/R insult, with and without rasagiline treatment, were permeabilized with Fix & Perm Cell Permeabilization Kit (Life Technologies, Carlsbad, CA, USA) for 30 min at 4 °C. Thereafter, the cells were immunostained with a mouse anti-GAPDH antibody (1:200–1:1000; Santa Cruz Biotechnology), mouse monoclonal antibody anti α-synuclein-1 (1:50; Transduction Laboratories), rabbit polyclonal anti α-synuclein (1:100; Chemicon), or mouse monoclonal anti-Nrf2 (1:100; Santa Cruz Biotechnology Inc.). The secondary antibodies used for immunofluorescence (1:1000–1:5000) were goat anti-mouse or anti-rabbit conjugated to Alexa Fluor 594 (red, Abcam, Cambridge, UK) or Alexa-Fluor-488 (green, Abcam). Nuclear staining was performed with DAPI diluted 1:1000 in PBS for 3 min (blue staining). The cells were examined by a fluorescent microscope (Nikon 50i. Nikon Instruments Inc. Melville, NY. USA) at magnifications of 100× or 200×. In other experiments, the coverslips were observed at a magnification of 600× using a confocal microscope (Zeiss, Oberkochen, Germany, LSM 710). Photographs of different random fields (4–5 per slide) were acquired, and 60–80 cells were analyzed for their fluorescence intensity (arbitrary units) using Image J software version number 1.51p. Representative fields were quantified and displayed [40]. 

### 2.10. Statistical Analysis

The results were statistically analyzed using SPSS 19.0 software (SPSS Inc., Chicago, IL, USA). The results are presented as mean ± SEM. Each experiment was performed four to six times in sixplicate wells. Comparisons between experimental groups were performed using the analysis of variance program (ANOVA), followed by Dunnett’s multiple comparison test. A *p*-value of 0.05 or less was considered significant for all comparisons. 

## 3. Results

### 3.1. Rasagiline Conferred Neuroprotection towards Ischemic Insult

#### 3.1.1. Rasagiline Inhibited Ischemic Insult-Induced Necrotic Cell Death

The ability of rasagiline to exert its neuroprotective effect was tested in a PC12 cell model exposed to OGD/R. Briefly, monolayer cultures of PC12 cells were exposed to 4 h of OGD followed by 18 h of reoxygenation in the presence of different micromolar concentrations of rasagiline [34] or 1500 μM tempol antioxidant, a concentration previously found most effective for neuroprotection [35] (Figure 1A). It is evident that the addition of 10 μM rasagiline significantly (*p* ≤ 0.01%) decreased necrotic cell death by 33% as measured by LDH release compared to control OGD/R-exposed but untreated cells. This cytoprotective effect indicates that 10 μM rasagiline conferred 80% neuroprotection, which was twofold higher compared to the effect of tempol (Figure 1A-insert).

#### 3.1.2. Rasagiline Inhibited Ischemic Insult-Induced ROS Production

To examine whether rasagiline treatment suppressed the ischemic insult-induced oxidative stress, the levels of mitochondrial ROS were measured in PC12 cell cultures exposed to ischemic insult (OGD/R) in the presence or absence of 10 μM rasagiline by comparison to normoxic cultures with and without rasagiline treatment (Figure 1B,C). The levels of ROS measured in the PC12 cell cultures exposed to ischemic insult in the absence or presence of 10 μM rasagiline were 185 ± 5% and 162 ± 4%, respectively, as compared to 100 ± 15% in the control normoxic cells with and without rasagiline treatment. Therefore, the ischemic insult-induced elevation of ROS in PC12 cell cultures was significantly decreased by about 15% upon rasagiline treatment (*p* ≤ 0.05, Figure 1B,C), as was also reported upon tempol treatment [35].

#### 3.1.3. Rasagiline Inhibited Ischemic Insult-Induced Apoptotic Cell Death

PC12 cell cultures exposed to OGD/R showed increased apoptosis, which significantly decreased in cell cultures treated with 10 μM rasagiline (Figure 1D). Ischemic insult exposure (OGD+) increased the activation of apoptotic effectors, including the cleavage of caspase-3 and PARP, while pretreatment with 10 μM rasagiline during the ischemic insult (OGD+/rasagiline+) significantly reduced the levels of these proteins by 60–80% and increased the expression of the anti-apoptotic protein Bcl-2 by 70% (upon comparing the protein level of expression relative to beta-actin levels). These results indicate that rasagiline inhibited ischemic insult-mediated apoptosis in the PC12 cell cultures.

### 3.2. Rasagiline Potentiated PI3K/Akt Signaling in PC12 Cell Cultures Exposed to Ischemic Insult

Activation of the PI3K/Akt signaling pathway by ischemic insult in PC12 cell cultures has been reported in the past [41]. To elucidate whether this intracellular signaling pathway could be involved in rasagiline-induced neuroprotection towards the ischemic insult, we investigated the degree of activation of Akt by the ischemic insult at the beginning of the reoxygenation phase in the presence and absence of rasagiline (Figure 2). 

The phosphorylation experiment indicated strong phosphorylation of Akt in cultures exposed to OGD and 30 min reoxygenation in the presence of 10 µM rasagiline, but this was lower by 50% in the untreated ischemic cultures (Figure 2; *p* < 0.01). Based on the results described above, we examined whether a PI3K inhibitor could inhibit rasagiline-induced PI3K phosphorylation. The PI3K inhibitor LY29004 inhibited ischemic insult-induced as well as rasagiline ischemic insult-induced phosphorylation of Akt (Figure 2). In normoxic cultures, the effects of rasagiline either alone or in the presence of LY294002 on the phosphorylation of Akt were non-significant. In parallel experiments, the effect of the PI3K inhibitor LY29004 on the neuroprotective effect of rasagiline towards the ischemic insult was investigated. It was found that LY29004 treatment reduced the rasagiline-induced neuroprotective effect significantly, by 36% (Table 1, *p* < 0.05), indicating a relationship between the partial reversal of rasagiline-induced Akt activation and the rasagiline-induced neuroprotective effect during the ischemic insult. This further indicates that additional mechanisms may contribute to neuroprotection. 

### 3.3. Rasagiline Inhibited the Ischemic Insult-Induced GAPDH Nuclear Translocation

GAPDH’s nuclear translocation participates in neuronal and non-neuronal cell death [42]. Therefore, we sought to investigate whether the ischemic insult could cause the translocation of GAPDH from the cytosol into the nucleus in PC12 cell cultures, either undifferentiated or differentiated with NGF, and whether rasagiline could modulate this effect (Figure 3). In the control group (normoxia), immunostaining indicated that the GAPDH level was low and was found predominantly in the cytosol (Figure 3A insert, red staining) in both undifferentiated and NGF-differentiated cell cultures. However, the GAPDH signal increased by 3–4-fold compared to the control upon exposure to the ischemic insult (Figure 3A; * *p* < 0.01 compared to control). The treatment with 10 μM rasagiline inhibited GAPDH immunostaining induced by the ischemic insult by 2.6-fold, with a higher intensity in the NGF-differentiated compared to the undifferentiated cell cultures (Figure 3A; * *p* < 0.01 compared to OGD). To confirm the quantitative results of confocal microscopy, GAPDH levels were determined by Western blotting to investigate the effect of OGD and rasagiline on the nuclear localization of GAPDH (Figure 3B). The results indicated that OGD/R induced an increase in the translocation from the cytosol to the nucleus of GAPDH, an effect which was attenuated by 75–90% by 10 µM rasagiline treatment in both undifferentiated and NGF-differentiated cells (Figure 3B). These results may suggest that rasagiline’s inhibition of ischemic insult-induced cell death may have been due to the attenuation of the translocation and transcriptional activity of GAPDH in the PC12 cell nuclei.

### 3.4. Rasagiline Increased Translocation of Nrf2 into the Nucleus and Transcription of ARE Phase II Antioxidant Enzyme Genes during the Ischemic Insult 

Considering that the transcription factor Nrf2 is a master regulator of the acute antioxidant defense [43], we hypothesized that rasagiline could modulate its expression and subcellular distribution during the ischemic oxidative insult. Immunostaining and Western blotting analyses were performed to determine the extent of nuclear translocation of Nrf2 in response to the ischemic insult with and without treatment with different concentrations of rasagiline (Figure 4). The fluorescence microscopy staining indicated that under normoxic conditions, Nrf2 has a low cytoplasmic distribution. This was increased by the ischemic insult and upon treatment with 10 µM rasagiline localized in part with the DAPI-blue stained nuclei (Figure 4A). Western blotting results showed that pretreatment with 1 and 5 µM rasagiline significantly increased the nuclear levels of Nrf2 by 40% and 90%, respectively (Figure 4B,C; *p* < 0.01). To further determine the role of Nrf2 in the mechanism of action of rasagiline, we used siRNA treatment to significantly reduce the expression level of Nrf2 in PC12 cells. The reduced level of Nrf2 was confirmed by Western blotting analysis (Figure 4B,C). In PC12 cells transfected with non-targeting scramble siRNA, no evidence of silencing was found because the Nrf2 levels were most probably similar to the control. By contrast, in the Nrf2 siRNA-transfected PC12 cells, the rasagiline-induced increase in the nuclear level of Nrf2 during the ischemic insult was reduced by about 60% (Figure 4B,C; *p* < 0.01).

In parallel experiments, the effect of the Nrf2-siRNA on the neuroprotective effect of rasagiline towards the ischemic insult was investigated. It was found that Nrf2-siRNA, but not scramble RNA, treatment reduced the rasagiline-induced neuroprotective effect by 70% (Table 1; *p* < 0.01), suggesting an apparent direct correlation during the ischemic insult between the rasagiline-induced increased translocation of Nrf2 into the nucleus and the rasagiline-induced neuroprotective effect. 

Increased levels of antioxidant enzymes such as NQO1, HO-1, catalase, etc., by the Nrf2 pathway provide a major defense against oxidative stress in Parkinson’s [44] and stroke [45]. To investigate whether rasagiline affected the expression of these antioxidant genes in PC12 cell cultures exposed to the ischemic insult, the mRNA expression levels of *NQO1*, *HO-1,* and *catalase* genes were determined (Figure 5). The ischemic insult increased the mRNA levels of *NQO1*, *HO-1*, and *catalase* by about twofold as compared to the normoxic cultures (Figure 5; *p* < 0.05). The increases in the mRNA levels of *NQO1*, *HO-1*, and *catalase* were amplified by 6.4- and 3.5-fold, respectively, upon treatment with rasagiline during the ischemic insult (Figure 5; *p* < 0.01), which is indicative of increased transcription of the ARE-mediated phase II detoxifying antioxidant enzyme genes.

### 3.5. Rasagiline Decreased the Expression Levels of α-Synuclein during Ischemic Insult 

Since the α-synuclein protein is involved in cell death and oxidative stress in PC12 cells [46], and since preventing its expression has a neuroprotective effect after stroke [25], we sought to investigate the effect of rasagiline on both undifferentiated and NGF-differentiated PC12 cell cultures exposed to the OGD/R ischemic insult (Figure 6). Undifferentiated and 50 ng/mL NGF-differentiated PC12 cell cultures grown under normoxic conditions or exposed to ischemic insult in the presence or absence of 10 µM rasagiline were fixed, permeabilized, and immunostained. The ischemic insult increased the level of α-synuclein expression in the ischemic cultures by about twofold as compared to the normoxic cultures (Figure 6; *p* < 0.05). This increased effect was attenuated in the ischemic cultures treated with rasagiline (*p* < 0.01, Figure 6A,B). The representative Western immunoblots and densitometry data for the α-synuclein protein levels of the undifferentiated and NGF-differentiated PC12 cell culture protein lysates indicated that both the monomeric (about 14–19 kDa) and tetrameric (about 56–76 kDa) forms of α-synuclein were significantly increased in the ischemic cultures compared to the normoxic cultures, and were reduced by about 50% in the ischemic cultures treated with 10 µM rasagiline (Figure 6B).

## 4. Discussion

Current major treatment strategies for PD and stroke are directed towards improving and/or reducing the symptoms of the disease without modifying its underlying multifactorial pathology [47,48]. Thus, the quest for efficient disease-modifying treatments (attempts to delay/slow progression by addressing the underlying pathology of the disease) with chronic therapeutic effects on disease progression has been envisioned as a novel approach, with many drugs undergoing active clinical trials [49,50]. Type-B monoamine oxidase inhibitors, such as rasagiline (Azilect) and selegiline (Deprenyl), treat Parkinson’s patients by ameliorating motor symptoms and improving motor fluctuations, and their evaluation in preclinical studies has indicated that they hold strong neuroprotective potential in Parkinson’s and other neurodegenerative diseases by reducing oxidative stress [51]. However, MAO-B inhibitors have been poorly characterized in vitro and in vivo for neuroprotection in ischemic stroke [52]. 

In the past, we investigated the in vitro neuroprotective features of chemicals, drugs, natural products, and stem cells on PC12 clonal cell lines temporarily deprived of oxygen and glucose (OGD), followed by reoxygenation (OGD/R insult). These catecholaminergic neurons have previously been used to mimic some of the properties of in vivo brain ischemia–reperfusion injury (IRI) and have been instrumental in identifying common mechanisms such as calcium overload, redox potential, lipid peroxidation, MAPKs modulation, etc. [33,53,54]. The present study characterizes several signaling pathways, which are beneficial in terms of the neuroprotective effects of rasagiline towards ischemia-like (OGD/R)-induced neuronal injury in vitro using the PC12 cell culture model, as is schematically presented in Figure 7. 

First, rasagiline, a selective MAO-B inhibitor, attenuated the ischemia-induced aponecrotic PC12 cell death by decreasing the generation of free radicals (Figure 7, pathways 1 and 3). Similar findings were observed in the brains of rodent models, in which rasagiline reduced dopamine oxidative metabolism via both MAO B inhibition and a direct, MAO-B-independent antioxidant effect [55,56,57].

Secondly, rasagiline decreased the nuclear translocation of GAPDH, thereby reducing cell death (Figure 7, pathway 2), as is expected from accumulated evidence demonstrating that GAPDH nuclear translocation plays a critical role in ischemic cell death [58]. In support of the present findings, there are studies indicating that GAPDH-MAO-B-mediated cell death, induced by different insults, is prevented by rasagiline [13,30,59] and selegiline [60,61,62] in different cell systems. A plausible mechanism by which rasagiline decreased the nuclear translocation of GAPDH is the activation of Akt during the ischemic insult (Figure 7, pathway 4), which in turn phosphorylated the Thr-237 of GAPDH and decreased its nuclear translocation, an essential step for GAPDH-mediated apoptosis.

Thirdly, rasagiline decreased ROS levels/oxidative stress and mitochondrial dysfunction (Figure 7, pathway 3), most probably through a direct antioxidant effect, since its propargylamine pharmacophore can directly scavenge free radicals [57]. Propargylamine residue is composed of an amine group in β-position to an alkyne moiety, and compounds with a carbon–carbon triple bond can behave as electrophilic substrates and as electron sources in nucleophilic reactions, providing a direct antioxidant effect. Other studies supporting Figure 7, pathway 3 indicate that selegiline inhibits NOS in the brain’s mitochondria, potentiates mitochondrial cytochrome oxidase activity, and reduces ROS/NOS production [63,64]. 

Fourth, rasagiline activated the PI3K-Akt-Nrf2 pathway in the PC12 cell cultures exposed to the ischemic insult with the induction of the transcription of the antioxidant response element (ARE) genes (Figure 7, pathway 4). In different oxidative stress cell type models, selegiline and rasagiline induced the nuclear translation of Nrf2; increased binding to the antioxidant response element (ARE); enhanced the expression of the antioxidant thioredoxin; and increased the activities of glutathione-dependent antioxidant enzymes, anti-peroxidative enzymes, catalase, and superoxide dismutase [65,66,67]. Cerebral ischemic stroke involves many pathological processes, such as oxidative stress, inflammation, and mitochondrial dysfunction. Nrf2, as one of the most critical antioxidant transcription factors in cells, can coordinate various cytoprotective antioxidant enzymes and factors to inhibit oxidative stress. Targeting Nrf2 is considered as a potential strategy to prevent and treat cerebral ischemic injury [45]; therefore, rasagiline may be repurposed for ischemic stroke therapy. 

Together, these events ultimately lowered the expression of the monomeric and tetramer-neurotoxic α-synuclein protein level (Figure 7, pathway 5), minimizing cell death. α-synuclein is a small, soluble, disordered protein that is widely expressed in the nervous system. Although its physiological functions are not yet fully understood, it is mainly involved in synaptic vesicle transport, neurotransmitter synthesis and release, cell membrane homeostasis, mitochondrial and lysosomal activities, etc. The complex pathological manifestations of α-synuclein are attributed to its structural complexity, misfolding, and different post-translational modifications. These properties cause mitochondrial dysfunction, oxidative stress, and neuroinflammation, resulting in neuronal cell death and neurodegeneration. Several recent studies have discovered the pathogenic roles of α-synuclein in traumatic and vascular central nervous system diseases, such as traumatic spinal cord injury, brain injury, and stroke, as well as in aggravating the processes of neurodegeneration [68]. The beneficial effect of rasagiline, i.e., lowering the expression of α-synuclein protein levels, further stresses its potential clinical use in therapy for synucleinopathic diseases [57].

The neuroprotective effects of rasagiline towards brain ischemia in vivo were also investigated in a rat model of ischemic stroke and a mouse model of traumatic brain injury. Rasagiline, in a dose regimen of 1–3 mg/kg delivered intraperitoneally within 16 h or by sustained intravenous infusion to maintain a 3 h steady state, improved the outcome of permanent middle cerebral artery occlusion (MCAO) in the rat stroke model. Rasagiline reduced the infarct size by 48.6% and the neurological score by 32.7%. Cognitive functions, tested in a water maze 2–3 weeks after occlusion, also significantly improved. The necrotic brain area was 35–50% smaller with rasagiline following a single bolus dose [69]. Rasagiline at doses of 0.2 and 1 mg/kg injected 5 min after closed-head traumatic brain injury in the mice accelerated the recovery of motor function. Daily injection of 1 mg/kg rasagiline from day 3 after injury improved spatial memory and reduced the cerebral edema by about 40–50% [31]. Rasagiline monotherapy has been established in early Parkinson’s disease (PD) for motor benefits in patients from both Eastern and Western countries, as is evident from a recent meta-analysis of randomized controlled clinical trials [70], and a longer duration of MAO-B inhibitor exposure is associated with reduced clinical decline in Parkinson’s disease [71]. In a phase II randomized, double-blind, and placebo-controlled study, selegiline treatment facilitated recovery after stroke [72]. However, as reported here, additional preclinical studies combined with necessary clinical studies are required in order to unambigously prove the concept of a disease-modifying, neuroprotective effect of MAO-B inhibitors in PD and stroke. 

## 5. Conclusions

Rasagiline (Azilect) is a selective MAO-B inhibitor drug that provides symptomatic benefits in PD treatment and has been found to exert neuroprotective effects in preclinical cellular and animal models of ischemic stroke. However, slowing or halting the neurodegenerative process has not yet been accomplished in PD or stroke patients using these drugs; therefore, neuroprotection is still considered an unmet clinical need. We investigated the neuroprotective signaling pathways of rasagiline in a PC12 dopaminergic neuronal cell culture model, which was exposed to oxygen–glucose deprivation (OGD) followed by reoxygenation (OGD/R, ischemic-like insult). Rasagiline decreased the production of neurotoxic reactive oxygen species and aponecrotic cell death, activated protein kinase B (Akt) phosphorylation activity, decreased the nuclear translocation of GAPDH, induced the nuclear shuttling of transcription factor Nrf2, and increased the mRNA expression of antioxidant enzymes in temporal relation to the neuroprotective effect. These results indicate that rasagiline conferred neuroprotection via improving mitochondrial integrity, as well as increasing mitochondria-specific antioxidant enzymes through a mechanism involving the Akt/Nrf2 redox-signaling pathway. The pathological effects of oxidative stress, mitochondrial dysfunction, and cell death induced by the OGD/R ischemic-like insult in the PC12 cell culture model were similarly induced by other neurotoxic chemical insults such as rotenone. Many reports have indicated that neuroprotectants alleviate rotenone-induced oxidative toxicity through an Nrf2/HO-1-mediated antioxidant and anti-apoptotic mechanisms [73], as was found in the present study with rasagiline. All these findings indicate a basic role of the Nrf2 signaling pathway in the neuroprotective activity against oxidative toxicity induced by different neurotoxins with distinct mechanisms of action, exerting an anti-Parkinsonian effect. These findings may be exploited to develop a third generation of MAO-B inhibitors with improved neuroprotection in PD and stroke disease-modifying therapy. However, further investigations are required in order to provide evidence for this concept in neuronal stroke models both in vitro and in vivo, as well as in clinical trials.

## Figures and Tables

**Figure 1 biomedicines-12-01592-f001:**
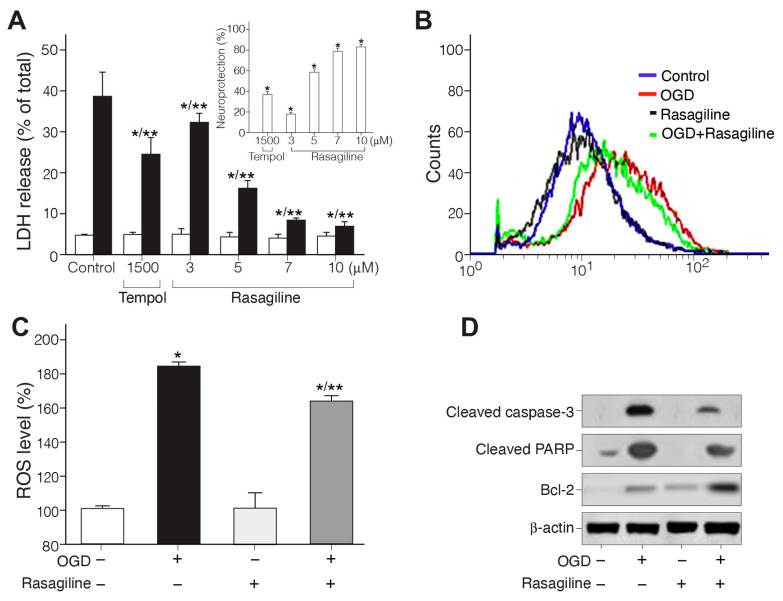
Rasagiline reduced aponecrotic cell death and the production of mitochondrial reactive oxygen species in PC12 cell cultures exposed to ischemic insult. (**A**) PC12 cells (1.2 × 10^6^ cells/well) were applied to 12-well plates and grown for 3 days. At the start of the experiment, the normoxic (white bar) and ischemic insult-exposed cell cultures (black bar) were treated with different concentrations of rasagiline or tempol (1500 μM). The OGD/R insult was carried out for 4 h, followed by 18 h of reperfusion. Aliquots from the culture media were taken for LDH release measurements, indicating that rasagiline decreased necrotic cell death. * *p* ≤ 0.01 vs. normoxia; ** *p* ≤ 0.01 vs. OGD, control; Insert: neuroprotection; * *p* ≤ 0.01 vs. control. (**B**) Mitochondrial reactive oxygen species (ROS) production was examined by flow cytometry, and typical traces are presented. (**C**) Quantitation of ROS indicating that the elevation of ROS level by the ischemic insult was significantly attenuated by treatment with rasagiline; * *p* ≤ 0.01 vs. control, normoxia; ** *p* ≤ 0.05 vs. OGD. (**D**) Western blots of cell culture extracts indicate that rasagiline treatment during the ischemic insult (OGD) inhibited the expression levels of apoptotic effectors (cleaved caspase-3 and cleaved PARP) and increased the levels of the anti-apoptotic protein Bcl-2.

**Figure 2 biomedicines-12-01592-f002:**
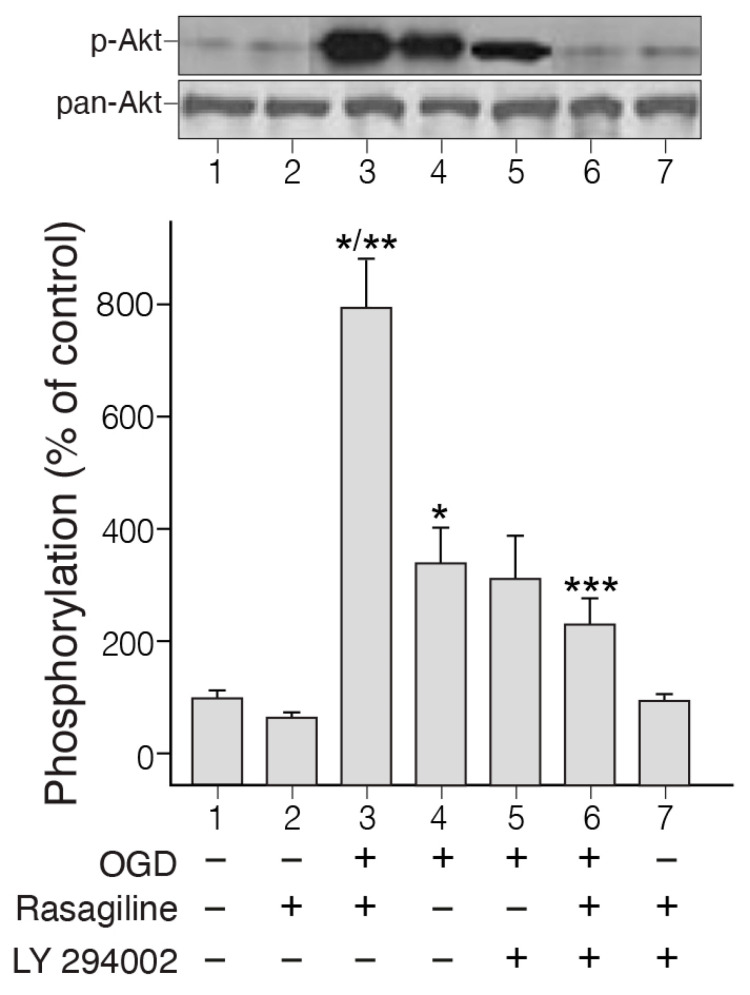
Rasagiline potentiates Akt signaling in PC12 cell cultures exposed to ischemic insult. PC12 cell cultures were subjected to 4 h of OGD followed by 30 min of reperfusion in the absence (lane 4) or presence of 10 μM rasagiline (lane 3), as well as with (lane 6) and without 50 μM LY29004 (lane 3). Normoxic cultures were subjected to drugs (lanes 2, 7) or left untreated (lane 1). Cell extracts were prepared for Western blot analyses of Akt phosphorylation (top blots) and total enzyme level using pan antibodies (bottom blots). The relative phosphorylation of the kinase was calculated by the ratio between the phosphorylated and total (pan) levels. Data are expressed as a percentage above control cells and represent the mean ± SEM (*n* = 4); * *p* < 0.01 vs. normoxia (lane 1); ** *p* < 0.05 vs. OGD alone (lane 4) or OGD and rasagiline (lane 3); *** *p* < 0.001.

**Figure 3 biomedicines-12-01592-f003:**
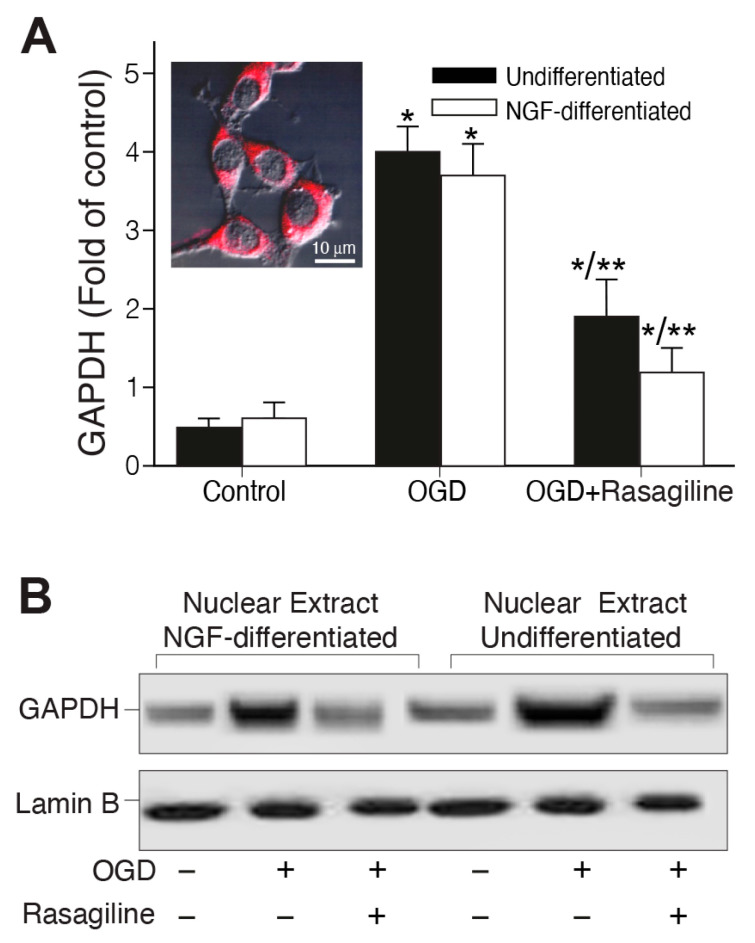
Rasagiline decreased ischemic insult-induced nuclear translocation of GAPDH. Undifferentiated (black bar) and NGF-differentiated (white bar). PC12 cell cultures were subjected to normoxia (control) or ischemic insult (OGD/R) in the absence (−) or presence (+) of 10 μM rasagiline (OGD + Rasagiline). (**A**) Quantitative analysis of the fluorescence intensity of cultures grown on coverslips, fixed and immunostained with anti-GAPDH antibody (red), and analyzed by a confocal microscope (insert), indicating that it wasmostly located in the cytosol and its levels were increased by the ischemic insult and reduced by rasagiline treatment. * *p* < 0.01 vs. control; ** *p* < 0.01 vs. OGD. (**B**) Western blotting analysis of the 36 kDa nuclear GAPDH protein; the nuclear protein of 66 kDa Lamin-B was used as control; GAPDH level was increased by the ischemic insult and decreased in rasagiline-treated cultures.

**Figure 4 biomedicines-12-01592-f004:**
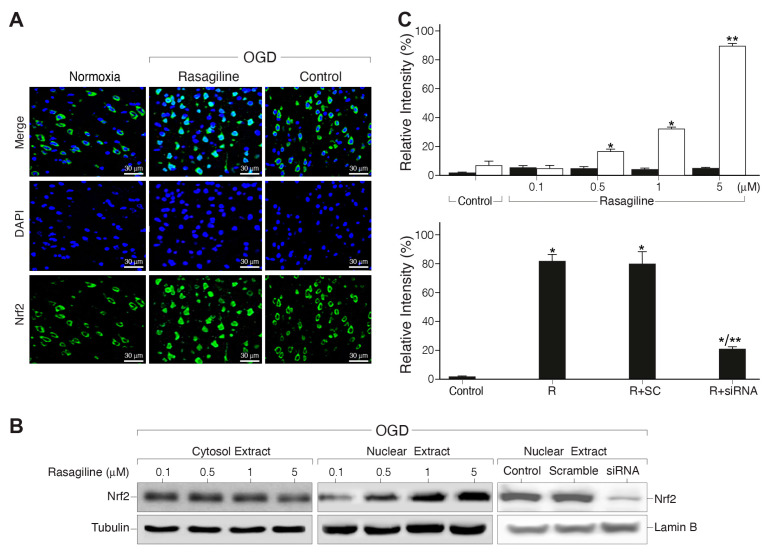
Rasagiline increased ischemic insult-induced nuclear translocation of Nrf2. PC12 cell cultures were subjected to normoxia or ischemic insult (OGD/R) in the absence (control) or presence of 10 μM (**A**) or different concentrations of rasagiline (**B**). (**A**) Representative immunofluorescence images of cultures grown on coverslips and fixed and stained with anti-Nrf2 antibody (green), as well as nuclei stained with DAPI and analyzed by a fluorescent microscope. Inserts: a few cells at higher magnification. (**B**) Representative images of the Western blotting analysis of cytosol (left); nuclear extracts treated during the ischemic insult (OGD) with different concentrations of rasagiline (middle); and nuclear extracts of untreated (control), scramble RNA (SC), and Nrf2-siRNA (siRNA) cultures (right). β-tubulin and Lamin B were used as controls for the cytosol and nucleus, respectively. (**C**) Quantitative analysis of the nuclear extracts presented in part (**B**). Rasagiline dose-dependently increased the nuclear Nrf2 protein level during the ischemic insult in naïve cells (white bar) compared to normoxia (black bar) (top), an effect that was inhibited in Nrf2-siRNA treated cultures (bottom). * *p* < 0.01 vs. control or respective normoxic level; ** *p* < 0.01 vs. rasagiline (R).

**Figure 5 biomedicines-12-01592-f005:**
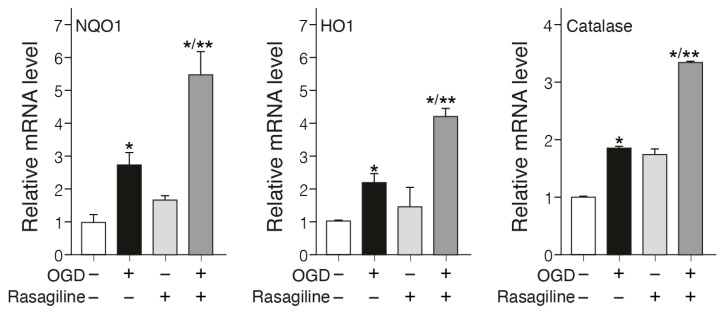
Rasagiline increased the mRNA levels of ARE-mediated expression of phase II antioxidant enzymes. PC12 cell cultures were subjected to normoxia or ischemic insult (OGD/R) in the absence or presence of 10 μM rasagiline, which significantly increased the mRNA levels of *NQO1*, *HO-1*, and *catalase* as compared to normoxic untreated or treated cultures. * *p* < 0.01 vs. normoxia; ** *p* < 0.01 vs. OGD.

**Figure 6 biomedicines-12-01592-f006:**
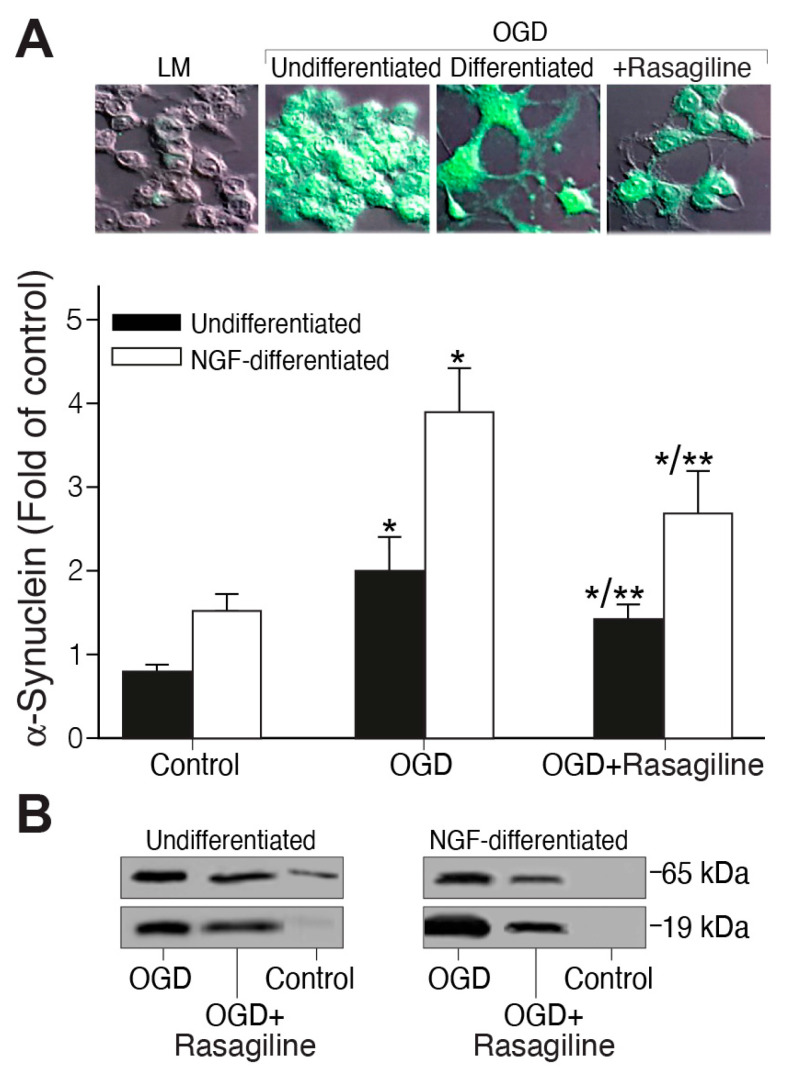
Rasagiline decreased the OGD/R-induced α-synuclein protein level. Undifferentiated (black bar) or NGF-differentiated (white bar) PC12 cell cultures were subjected to normoxia (control) or ischemic insult (OGD) in the absence or presence of 10 μM rasagiline (OGD + Rasagiline). (**A**) Quantitative analysis of the fluorescence intensity of cultures grown on coverslips, fixed and immunostained with anti-α-synuclein antibody (green), and analyzed by a confocal microscope. Its level was increased by the ischemic insult and reduced by rasagiline treatment. * *p* < 0.05 vs. control; ** *p* < 0.05 vs. OGD. LM—light microscopy photos of undifferentiated cells; (**B**) Western blotting analysis of the 19 and 65 kDa α-synuclein proteins. Their level was increased by the ischemic insult and significantly decreased in the rasagiline-treated cultures.

**Figure 7 biomedicines-12-01592-f007:**
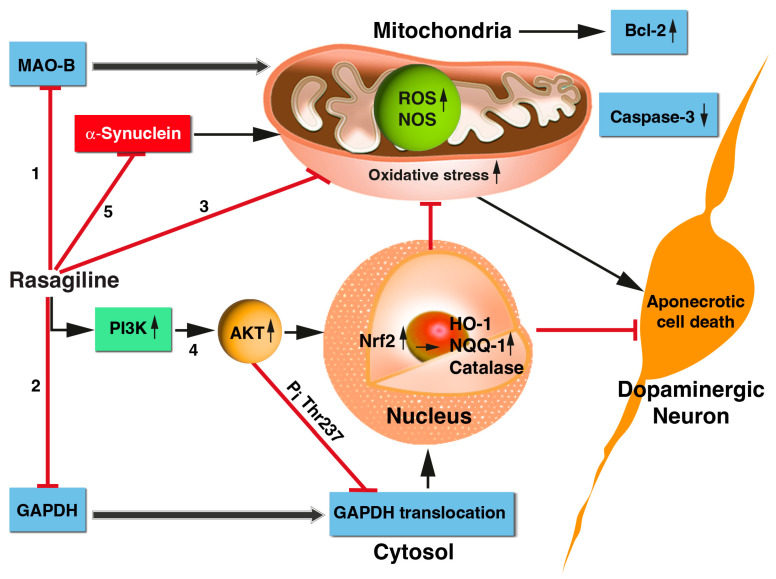
Summary of the underlying mechanisms by which rasagiline confers neuroprotection towards ischemic-like (OGD/R) insults of PC12 cell cultures. Rasagiline, a selective MAO-B inhibitor, attenuated the ischemia-induced neuronal injury through the following actions: 1. Inhibition of MAO B activity of metabolizing dopamine, therefore reducing the generation of free radicals during ischemia/reperfusion; 2. decreasing the nuclear translocation of GAPDH and thereby reducing cell death; 3. decreasing the ROS levels/oxidative stress and mitochondrial dysfunction via a direct antioxidant effect, since its propargylamine pharmacophore can directly scavenge free radicals; 4. activation of the PI3K-Akt-Nrf2 pathway and induction of the antioxidant response element (ARE) genes. 5. Together, these events and others ultimately lowered the expression of the monomeric and/or tetramer neurotoxic α-synuclein protein level, minimizing cell death. Black, sharp arrow: activate; red, blunt arrow: inhibit. Abbreviations: MAO-B, monoamine oxidase B; GAPDH, glyceraldehyde 3-phosphate dehydrogenase; Akt, protein kinase B; Bcl2, B-cell lymphoma 2 protein; Nrf2, nuclear factor erythroid 2-related factor 2; *HO-1*, *heme oxygenase 1gene*; *NQQ1*, *NAD(P)H quinone oxidoreductase 1gene*; ROS, reactive oxygen species.

**Table 1 biomedicines-12-01592-t001:** The effect of different inhibitory chemicals on the rasagiline-induced neuroprotection towards ischemic-like insult in PC12 cell cultures.

Treatment	LDH Release (% of Total)		Neuroprotection (% of Rasagiline)
	OGD	OGD + Rasagiline	
Control ^1^	38 ± 9	10 ± 3	74 ± 7
LY294002 ^1^	42 ± 7	22 ± 6 ^1,^*	48 ± 9
Scramble RNA ^2^	47 ± 5	15 ± 5 ^2^	68 ± 6
Nrf2 siRNA ^2^	36 ± 10	28 ± 6 ^2,^*	20 ± 4

^1^ PC12 cell cultures were subjected to 4 h OGD followed by 30 min reperfusion in the absence (OGD) or presence (OGD + Rasagiline) of 10 μM rasagiline with and without 50 μM LY29004. * *p* < 0.01 compared to control. Under normoxia, LDH release was 2% of total. ^2^ PC12 cell cultures treated for 5 days with scramble RNA or Nrf2-siRNA were subjected to the ischemic insult in the absence (OGD) or presence (OGD + Rasagiline) of 10 μM rasagiline. The OGD/R insult was carried out for 4 h, followed by 18 h reperfusion. Aliquots from the culture media were taken for LDH release measurements. * *p* < 0.01 compared to control or scramble RNA.

## Data Availability

All data associated with this study are available in the main text or are available through the corresponding author upon request due to privacy.

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
