# Peer review of "Rasagiline Exerts Neuroprotection towards Oxygen–Glucose-Deprivation/Reoxygenation-Induced GAPDH-Mediated Cell Death by Activating Akt/Nrf2 Signaling"

_biomedicines, 2024, doi:10.3390/biomedicines12071592_

Round 1

Reviewer 1 Report

Comments and Suggestions for Authors

The authors investigated the possible pathways involved in the protective effect of rasagiline in neuronal cells following an ischemia protocol in vitro. Based on the results, the study suggests that the drug could be beneficial for the treatment of Parkinson’s disease (PD). The significance of the content and findings presented in the manuscript will undoubtedly contribute to future research aimed at identifying new treatment strategies for PD.

After reviewing the document, I have some minor concerns. Firstly, the title is too lengthy and lacks clarity in its current form. In the abstract, the authors should disclose the concentrations of the drugs and compounds used, as well as the treatment schedule. Additionally, the methodology should include the NM accession numbers and amplicon lengths for the investigated genes.

It is worth noting that rotenone is commonly used as a model for Parkinson’s Disease in many studies, but this is not the case here. It would be beneficial to include more information on this topic in the text, perhaps in the Discussion section. Lastly, please carefully check abbreviations and ensure they are used consistently throughout the document.

Author Response

Please see the atachment

Reviewer 2 Report

Comments and Suggestions for Authors

In this manuscript titled, “Rasagiline confers neuroprotection to PC12 cell cultures exposed to oxygen-glucose deprivation/reoxygenation insult by activating the Akt/Nrf2 redox-signaling pathway and by decreasing the nuclear translocation of glyceraldehyde-3-phosphate dehydrogenase and α-synuclein expression”, the authors aimed to investigate the neuroprotective signaling pathways of rasagiline in the pheochromocytoma PC12 dopaminergic neuronal cell cultures exposed to an ischemic-like insult. In addition, the authors also explore the temporal effects of Akt activation and GAPDH nuclear localization in relation to rasagiline-induced neuroprotection. The authors claim that exposure of neurons to oxygen-glucose deprivation caused 40% cell death. Treatment with rasagiline induced a dose-dependent neuroprotection by increased Akt phosphorylation, decreased protein expression of the ischemia-induced α-synuclein protein and reactive oxygen species. Moreover, treatment with rasagiline induced nuclear shuttling of transcription factor Nrf2 and increased the mRNA levels of the antioxidant enzymes heme oxygenase-1, (NAD (P) H- quinone dehydrogenase, and catalase. These results of the study indicated that rasagiline conferred neuroprotection via improving mitochondrial integrity, as well as increasing mitochondria-specific antioxidant enzymes by a mechanism involving the Akt/Nrf2 redox-signaling pathway. The study is well designed and planned. Given the current treatment strategies for PD and stroke, that only reduce the disease symptoms rather than modifying the underlying pathology, this work may be of some importance that may add to the future drug development and repurposing for some aspects of PD and stroke therapy.

Concerns:

·         While the study exhibits some promise, conducting an in-vivo study on rodents would make the study more robust.

Reviewer 3 Report

Comments and Suggestions for Authors

This study demonstrated rasagiline as neuroprotection in the ischemic neuronal cultures with inhibition of α-synuclein and GAPDH-mediated aponecrotic cell death, and via mitochondrial protection, as by increasing mitochondria-specific antioxidant enzymes by a mechanism involving the Akt/Nrf2 redox-signaling pathway. Here, we have some general comments. First, although I found the research is organized well, the length of the title should be reduced. The title of research should be concise. Second, the figure 7 should be move to introduction part because it provide the relationship between Rasagiline, the Akt/Nrf2 redox-signaling pathway, glyceraldehyde-3-phosphate dehydrogenase and α-synuclein. Third, I have been seen the paper “The Protective Effect of DiDang Tang Against AlCl3-Induced Oxidative Stress and Apoptosis in PC12 Cells Through the Activation of SIRT1-Mediated Akt/Nrf2/HO-1 Pathway (doi: https://doi.org/10.3389/fphar.2020.00466). Please provide the novelty of your research in comparison with previous publications. In conclusion, the paper quite good. It should be accepted for publication in Biomedicines.
